# Have Lifestyle Habits and Psychological Well-Being Changed among Adolescents and Medical Students Due to COVID-19 Lockdown in Croatia?

**DOI:** 10.3390/nu13010097

**Published:** 2020-12-30

**Authors:** Ružica Dragun, Nikolina Nika Veček, Mario Marendić, Ajka Pribisalić, Gabrijela Đivić, Hellas Cena, Ozren Polašek, Ivana Kolčić

**Affiliations:** 1University Postgraduate Doctoral Study Program Evidence-Based Medicine, University of Split School of Medicine, 21000 Split, Croatia; ruzica.dragun@yahoo.com (R.D.); veceknika@gmail.com (N.N.V.); 2University Department for Health Studies, University of Split, 21000 Split, Croatia; mario_marendic@yahoo.com; 3Department of Public Health, University of Split School of Medicine, 21000 Split, Croatia; ajka.pribisalic@mefst.hr (A.P.); opolasek@gmail.com (O.P.); 4ENT Clinic, University Clinical Hospital Mostar, Bijeli Brijeg B.B., 88000 Mostar, Bosnia and Herzegovina; gabrijeladvc@gmail.com; 5Dietetics and Clinical Nutrition Laboratory, Department of Public Health, Experimental and Forensic Medicine, University of Pavia, 27100 Pavia, Italy; hellas.cena@unipv.it; 6Clinical Nutrition and Dietetics Service, Unit of Internal Medicine and Endocrinology, ICS Maugeri IRCCS, University of Pavia, 27100 Pavia, Italy

**Keywords:** lifestyle, Mediterranean diet, sedentary behavior, sleep, physiological stress, adolescent health, medical students, life change events, COVID-19, epidemic

## Abstract

Objective: To investigate dietary habits, sleep and psychological well-being of adolescents and medical students during COVID-19 lockdown in Split, Croatia. Methods: We surveyed 1326 students during 2018 and 2019, and compared their responses with 531 students enrolled in May 2020. Perceived stress, quality of life (QoL), happiness, anxiety, and optimism were assessed as proxies of psychological well-being, using general linear modelling. Results: We found no substantial differences in dietary pattern between pre-lockdown and lockdown periods, including the overall Mediterranean diet (MD) adherence. However, the MD pattern changed, showing increased adherence to the MD pyramid for fruit, legumes, fish, and sweets, while cereals, nuts, and dairy intake decreased during COVID-19 lockdown. A third of students reported weight loss during lockdown, 19% reported weight gain, while physical activity remained rather stable. The most prominent change was feeling refreshed after a night’s sleep, reported by 31.5% of students during lockdown vs. 8.5% before; median length of sleep duration increased by 1.5 h. Lockdown significantly affected QoL, happiness, optimism (all *p* < 0.001), and perceived stress in students (*p* = 0.005). MD adherence was positively correlated with QoL and study time, and negatively with TV and mobile phone use in pre-lockdown period (all *p* < 0.001). Interestingly, higher MD adherence was correlated with less perceived hardship and greater happiness and QoL during lockdown. Conclusion: These insights provide valuable information for tailored interventions aimed at maintaining healthy lifestyle in young population. Given the numerous beneficial effects associated with MD adherence, modification of lifestyle through application of lifestyle medicine deserves a priority approach.

## 1. Introduction

COVID-19 pandemic, caused by the novel coronavirus (SARS-CoV-2), is an unprecedented disease with tremendous impact on all of the humanity, with long-term consequences that still need to be uncovered. The virus spread rapidly and globally in a very short period of time causing panic and resulting in enforced restrictions by public health authorities in many countries in the world, including travel bans, restricting social gatherings and closing public schools [1]. Different countries have had differing levels of restrictions for the population, and during spring 2020 Croatia was one of the countries with the strictest measures on global scale [2]. Shortly, the first case of COVID-19 in Croatia was registered on 25 February 2020, and first measures were introduced on March 9, while schools and universities switched to a full-online learning on 16 March [3]. Few days later all entertainment and other public venues were closed, including restaurants, movie theatres, gyms, shopping centers and places of worship, culminating in a full lockdown on 23 March, banning free movement of people across countries without a special authorization [3]. After lockdown phase of 30 days, easing of restrictions began on 27 April. On 11 May preschool institutions (kindergartens) and schools were reopened, but only for the youngest children in the primary education (first to fourth grade); everyone else was left in online-teaching mode until the end of the school year in June [3].

Inevitably, those and similar measures tremendously impacted economy and functioning of society at large [4], but they also affected everyday life and health behaviors [5]. Unavoidably, restrictions on travel and outdoor activities, along with physical distancing, self-isolation and quarantine, disarranged routine daily activities, affecting in particular physical activity and eating habits of all age groups [5,6,7,8]. These new conditions and the overall uncertainty resulted in pronounced distress, with a negative impact on psychological well-being [9], with increased reported depression and anxiety symptoms throughout the population [10]. In return, those mental health disturbances have the potential to instigate unhealthy behaviors as coping mechanisms [7,11].

Health-related lifestyle habits develop in childhood and are reinforced in adolescence. They impact human health to a great extent in later life [12], making early stage of life particularly important for adoption and maintenance of healthy habits. Young people are more open to embracing novelty and changes, which in given situations can be a positive trait, but they are also more vulnerable to sudden changes. The period of intense studying requires a lot of physical and mental energy as the students have to divide their time between lectures, study and exams, possibly work and social life.

Young adults transitioning from secondary education to university level education, experience profound changes in their lives, including common nighttime sleep disruption [13], with decreased sleep quality caused by stress and anxiety [14]. Additionally, dietary habits may change drastically, with common barriers to healthy eating, primarily due to lack of time, unhealthy snacking and use of energy-dense comfort foods, increased stress, higher prices of healthy food, and easy access to junk food [15].

On top of the famed challenging life stage transitioning to adulthood, COVID-19 pandemic has drastically changed lifestyle and educational landscape of students at all levels of education, with unprecedented consequences. For example, various effects have been recorded among undergraduate students at Arizona State University (USA), with systematical differences according to their socioeconomic status and health, resulting in either a decrease or increase of the time spent studying, graduation delay, job losses and job offers acceptance reduction, as well as internship interruptions [16].

Compared to domestic students, international students have faced even worse experience during COVID-19 lockdown, separated from their families, friends, and familiar environments due to travel ban and border restrictions. This situation, which lasted for several weeks in most countries, may have enforced a significant strain on their coping mechanisms and lifestyle habits.

As of recently, more studies have been published on COVID-19 lockdown effects on dietary habits, as well as on other lifestyle behaviors [5,6,7,17,18]. However, most of the previous studies focused on adult population, and only few studies have been published so far about COVID-19 lockdown influence on lifestyle change in adolescents [19,20,21] and in medical students [22,23]. Hence, we aimed to investigate dietary habits and other related lifestyle behaviors changes in adolescents and medical students resulting to COVID-19 lockdown in Croatia. Given the uncertain course of COVID-19 epidemic in the future, such insights provide valuable information to the public health authorities for tailored interventions aimed at maintaining healthy lifestyle and preserving the health of the population for years to come.

## 2. Materials and Methods

### 2.1. Subjects

In this cross-sectional study we have included three groups of subjects in two periods: pre-COVID-19 lockdown (pre-COVID-19) and COVID-19 lockdown. All eligible students (the whole generation of students) were invited to participate. The first group of subjects consisted of secondary school students surveyed during their third year (out of four years). We included students enrolled in two consecutive academic years in pre-COVID-19 period; hence, two generations of secondary school students attending three gymnasium types of secondary schools in Split, second largest city in Croatia, during April-May 2018, and during the same period in 2019. We have included three out of six such schools in the city of Split, because three schools have been able to participate in data collection during lockdown period, while the remaining three were not able to organize participation of the whole generation of students during that challenging period and were excluded from this study. The overall response rate for secondary school students was 89.5% (*n* = 376) in 2018 and 93.6% in 2019 (*n* = 393).

Additionally, two groups of medical students from the University of Split School of Medicine were included in this study. Domestic medical students were Croatian citizens, studying in Croatian language, surveyed in their first, fifth and sixth year (out of six years of Medical school). International medical students, studying medicine in English language and mostly non-Croatian citizens (mainly coming from Germany, Scandinavian countries, UK, USA and Australia), were surveyed during the same years of Medical school. Overall response rate for domestic students, after inviting all eligible students was 81.8% (*n* = 207) in 2018 and 86.2% in 2019 (*n* = 206), while for international students response rates were 62.8% (*n* = 76) in 2018 and 58.1% in 2019 (*n* = 68).

The second study period took place in May 2020, right after the beginning of the decline of the restrictive measures. New generation of students attending the same schools and study years were surveyed, yielding a sample of 326 secondary school students (response rate of 77.1%), 149 domestic medical students (58.4%), and 59 international medical students (67.0%).

The survey was anonymous, and before providing their responses, all students were informed on the aim of the study, and the purpose of data collection. Only students who were younger than 18 years had their parents invited to sign the informed consent before answering the questionnaire. Students who were ≥18 years old were not asked to sign informed consent. They were informed that they were giving their consent at the time of filling anonymously the questionnaire and subsequently participating in the study. Data collection among secondary school students was organized within the science popularization project “Science on the plate: the foods of the Mediterranean” supported by the Ministry of Science and Education of the Republic of Croatia. Both groups of medical students were included within the HOLISTic study (Habits, Orthorexia Nervosa and Lifestyle in Students; ClinicalTrials.gov Identifier: NCT04252924). The study was conducted in accordance with the Declaration of Helsinki, and was approved by Ethics Committee of the University of Split School of Medicine (2181-198-03-04-18-0027).

### 2.2. Questionnaire

The anonymous questionnaire was self-administered in a paper-and-pencil form during 2018 and 2019, while in 2020 this was switched to an online data collection approach for safety issues and because students had only online classes at that time. In order to increase the response rate in all instances of data collection, we asked students’ representatives and professors for their help in informing students about this survey during their regular obligatory teaching hours (the whole generation was attending as a default). Students were asked to fill in the questionnaire at their earliest convenience.

The questionnaire had several sections, and it took on average 20 min to fill it out. The general section included questions on age, gender, school/study details (name of the school and study year), smoking habits (possible answers were yes; ex-smoker; never smoked), student’s weight, height, and how many days ago they had weighted themselves. Students were also asked to rate their health perception (Likert scale, values ranged between 0 and 10, where 0 meant very sick, and 10 represented complete health).

The dietary habits section of the questionnaire included questions on usual breakfast frequency during the week, number of main meals and snacks on working days and non-working days, and snacking habit while watching TV or studying (possible answers: yes, frequently; yes, sometimes; no). A validated questionnaire was used to assess the Mediterranean diet (MD) adherence [24], which was previously used in Croatian population [25,26,27], and validated in Croatian language [28]. Shortly, the Mediterranean Diet Serving Score (MDSS) combines 14 food groups, in accordance to the modern Mediterranean diet pyramid [29], including consumption of cereals, fruits, vegetables, olive oil, nuts, milk and dairy, potatoes, fish, red meat, white meat, eggs, legumes, sweets and wine (not included for underage students). Based on the frequency of those foods consumption, the maximum number of points on the MDSS score is 24, the originally proposed cut-off for MD adherence is 13.5, and we used a cut-off value of 14 points for students who were older than 18 years to denote students compliant with the MD. We excluded wine from scoring scheme for underage students, hence applying the cut-off value of 13 points. For details on scoring, please see the work by Monteagudo C. et al. and Kolčić I. et al. [24,25]. Besides MD food groups we also asked students about their intake of sweetened drinks and processed meat and fish (possible answers were: each day, twice or more a day; each day, once a day; 3 times a week; 2 times a week; once a week; once a month; rarely or never).

The section on sleeping habits included questions on bed-time as well as time of awakening (used to calculate sleep duration and sleep schedule) on both working days and non-working days (during COVID-lockdown we did not distinguish due to usual routine disruption and online learning instead of attending classes). Additionally, we asked students how they usually felt after waking up, with three possible answers: refreshed, tired and sleepy to a small extent, or extremely tired and sleepy.

The physical activity section of the questionnaire included questions on sedentary types of activities and frequency of active physical activity engagement. Sedentary types of activity included daily TV watching time (h/day), computer/tablet use time (h/day), mobile use time (h/day), daily study time (h/day), and finally, average daily sitting time (h/day). Assessment of active engagement in physical activities during pre-COVID-19 lockdown study period included questions on playing any sports, going to the gym and engaging in any other type of recreation, such as dance, yoga, Pilates, etc. (possible answers were: yes, a few times a week; yes, once per week; rarely; no). These three questions were included in the questionnaire in 2019, but they were not present in 2018, yielding a smaller sample size compared to other variables (660 students in total). Based on the combination of the answers on these three questions, students were classified into three categories of physical activity. Students who were active several times a week in any of the three types of activities, or in their combination, were considered to be weekly physically active. Students who responded they were active only once a week in only one of the three activity types, were considered to be active sometimes, while those students who answered they engaged rarely or never in any of those activities were considered to be active rarely or never. During the period of COVID-19 lockdown all the gyms were shut down and all programmed physical activities and sports were banned. Hence, we asked about students activity in general, e.g., walking, running, cycling, etc. (possible answers being: active every day of the week; active a few days a week; sometimes moderately active; very low level of activity and rarely active; not physically active at all).

Psychological wellbeing was assessed exploring several dimensions, such as perceived stress, quality of life (QoL), happiness, anxiety, and optimism about future. Stress level was evaluated by means of the Perceived Stress Scale (PSS-10) questionnaire, a validated and widely used questionnaire, assessing general perception of stressful situations during the last month. PSS-10 score ranges between 0 and 40 points, and as the score increases the perception of stress raises [30]. We also applied previously proposed cut-off values to identify students who had low level of stress perception (0–13 points), moderate (14–26 points) and high level of perceived stress (27–40 points) [31], even though dichotomization of PSS-10 score is not the usual approach [30].

Happiness, anxiety, and optimism about future were assessed using one question for each of those traits, and a Likert scale ranging between 0 and 10. Students had to choose one number for each question, which closely described how they felt, where “zero” meant not at all happy/anxious/optimistic, and “ten” represented extreme happiness/anxiety/optimism.

The questionnaire used for lifestyle habits assessment during COVID-19 lockdown specifically required students to refer to their habits in the two months from the very beginning of the lockdown. Additionally, the questionnaire used in 2020 included several new questions, specifically asking students to estimate changes in their usual behaviors during COVID-19 lockdown, compared to the period before lockdown. The questions included consumption of fruits, vegetables, meat and processed meat, sweets and snacks, change in body weight, screen time, including daily watching TV, use of computer/tablet, mobile phones, as well as daily study time (possible answers to all these questions were: reduced, remained the same, increased). Finally, students were asked to rate how hard their own experiences of lockdown and/or isolation had been, using the Likert scale ranging from zero to ten, where 0 corresponded to “not hard at all”, and 10 to “very hard”.

### 2.3. Statistical Analysis

Data were presented as absolute numbers and percentages for categorical variables, and medians with interquartile range (IQR) were used to describe ordinal variables and numerical variables, which did not follow normal distribution (tested with Kolmogorov-Smirnov test).

The difference between groups was tested using chi-square test, Mann-Whitney *U* test (for comparison of two groups) and Kruskal-Wallis test (for three groups).

Correlation between ordinal and numerical variables was performed using Spearman’s rank correlation test. Additionally, we created a multivariate general linear model to estimate the association of the lifestyle habits and COVID-19 lockdown with the psychological well-being in students (outcome variable). We entered all five psychological variables simultaneously as dependent variables (perceived stress score, quality of life, happiness, anxiousness, and feeling optimistic about future). We included the most important lifestyle habits as predictor variables in the model (Mediterranean diet, sleep duration, and sitting time as a measure of sedentary type of physical activity), alongside with time period (pre-COVID-19 was the referent point). All of these variables, except study period, were entered into the model as numerical variables. The model was also adjusted for the effect of confounding variables; age (numerical variable), gender (males were the referent group), and study group (secondary school students were the referent group). Using this model, we obtained adjusted *R*^2^ value of 11.5% for perceived stress score, 16.9% for the quality of life, 8.0% for happiness, 6.8% for anxiousness, and 5.9% for the optimistic feeling about future. We also reported the partial eta squared values for the model, in order to estimate effect sizes values, which indicate the percentage of variance in each of the effects and its associated error accounted for by that effect.

Statistical analysis was performed using the IBM SPSS Statistics software (v21.0; IBM, Armonk, NY, USA). Statistical significance was set at *p* < 0.05.

## 3. Results

A total of 1326 students sampled during pre-COVID-19 study period and 531 students during COVID-19 lockdown period were included in the analysis. We excluded one domestic medical student and two secondary school students from the analysis due to incomplete questionnaire response during COVID-19 lockdown study period. General characteristics of the students in two study periods and according to three study sub-groups are shown in Table 1. Females were roughly 2/3 of all three sub-samples. Secondary school students were on average younger than medical ones, and they had lower BMIs. There was no difference in smoking habits between study groups in either pre-COVID-19 or COVID-19 lockdown period, and the prevalence of active smokers ranged between 14.2% and 25.4%. Self-rated health perception was high in all study groups, while statistically significantly lower in international medical students during COVID-19 lockdown (median 8.0, IQR 2.0), compared to domestic medical students (median 9.0, IQR 2.0) and secondary school students (median 9.0, IQR 2.0; *p* < 0.001) (Table 1).

Comparison of lifestyle habits among all students before and during COVID-19 lockdown period is shown in Table 2. Most pronounced change was recorded in sleep duration and the way students felt after waking up. For instance, median sleep duration on working days during pre-COVID-19 lockdown was 7.0 h (IQR 1.5), while during COVID-19 lockdown period students reported to sleep more, a median of 8.5 h (IQR 1.5). As many as 30.8% and 10.2% of students reported to feel extremely tired and sleepy after waking up on working days during pre-COVID-19 lockdown and COVID-19 lockdown, respectively (*p* < 0.001; Table 2). 

Furthermore, we found no change in breakfast frequency (*p* = 0.350), snacking habits (*p* = 0.341) or in the overall MD adherence score between these two study periods (*p* = 0.927) (Table 2). During the COVID-19 lockdown period 9.2% of all students were compliant with the MD, compared to 12.1% in the period before COVID-19 (*p* = 0.077). During lockdown students reported slightly higher MD adherence for fruit (65.3% vs. 58.6%), legumes (60.6% vs. 53.3%), fish (32.8% vs. 24.4%), and sweets (30.5% vs. 22.4%) (Figure 1). On the other hand, cereals (24.1% vs. 35.6%), nuts (15.1% vs. 18.9%), and dairy products consumption was lower than in the pre-COVID period (22.4% vs. 31.8%) (Figure 1). Similar results were obtained in the sub-groups, with the most prominent overall differences recorded in international medical students, who showed higher adherence to most of the MD food groups during pre-COVID-19 period, unlike lockdown period (Appendix A). Those finding were partially confirmed in the sub-analysis of subjects surveyed during lockdown (Table 3). Ten percent of international medical students reported to have experienced a decrease of fruit and vegetables intake during lockdown, compared to their usual pre-COVID-19 habits, while only 1–5% of secondary school students and domestic medical students reported the same reduction (Table 3). Still, among subjects sampled during COVID-19 lockdown, 60–70% of students perceived they kept the same eating habits, while around 20–38% of them reported an increased intake of fruit and vegetables. However the same percentage of students showed increased consumption of sweets and snacks as compared to pre-COVID-19 lockdown period (Table 3). On the other hand, additional 20–30% of students reported they reduced consumption of sweets and snacks during lockdown. Finally, 30–40% of students reported body weight loss during lockdown, while around 20% of students observed weight gain (Table 3).

The most prominent difference in sedentary activity corresponded to the time spent on computer/tablet. This kind of activity was reported to average three hours per day during COVID-19 lockdown, which is two hours more than before due to the online learning (*p* < 0.001) (Table 2). As many as 78.4% of secondary school students increased their computer time during lockdown period (Table 3). Additionally, the majority of students sampled during lockdown also reported increased mobile phone use (Table 3). Even though screen time increased, average sitting time was actually shorter during lockdown in the overall sample of students (*p* < 0.001) (Table 2). When the sample was analyzed in subgroups, the highest duration of daily sitting time was reported by international students during COVID-19 lockdown (median of 10 sitting hours, IQR 3.5), while high school students reported less average sitting hours during lockdown (5 h [IQR 5.0] vs. 7.5 h [4.0] during pre-COVID-19 period) (Appendix A). 

More than half of students reported to be weekly physically active in both study periods, while a third of domestic medical students reported to be active rarely or never during the pre-COVID-19 period, unlike 9.0% of international medical students (Appendix A). One quarter of secondary school students reported they were active every day during lockdown, the same as 20.3% of domestic medical students and 13.6% of international ones (Table 3).

There was a difference in perceived happiness, optimism for future, anxiousness, and perceived stress between study periods, and students reported lower average quality of life during COVID-19 lockdown compared to pre-COVID-19 period (*p* < 0.001; Table 2). Additionally, high level of stress perception was reported by a higher percentage of subjects in both medical students groups during COVID-19 lockdown (Appendix A). However, three subgroups of students did not differ regarding their experience of hardship and isolation during lockdown, which was rated as being on average slightly higher than moderate (*p* = 0.623) (Table 3).

The correlation analysis revealed a positive correlation between perceived stress and anxiousness in both study periods, and those two characteristics were negatively correlated with happiness, being optimistic about the future and quality of life (Appendix A). Furthermore, subjective health rating was negatively correlated with daily sitting time, stress score, and anxiousness, while it showed a positive correlation with the MD adherence, sleep duration, quality of life, happiness and optimism during both study periods (Appendix A). MDSS score was slightly and positively correlated with quality of life in both periods and with happiness during lockdown period, while it was in slight negative correlation with hardship during lockdown (Appendix A). Sleep duration during working days was correlated with various sedentary types of physical activity and psychological characteristics (except with the quality of life) during pre-COVID-19 period. Mobile phone time was positively correlated with stress, sitting time and anxiousness during pre-COVID-19 (Appendix A), and with stress, anxiousness and hardship during lockdown (Appendix A), while it was negatively correlated with quality of life and happiness in both study periods.

Finally, we assessed the association between Mediterranean diet, sleep duration, sedentary activity, and COVID-19 lockdown with the overall psychological well-being, (Table 4). When we took into consideration gender, age, and study group as confounding variables, we found that MD was associated with quality of life (*F* = 14.95; *p* < 0.001), and optimism for the future (*F* = 7.23; *p* = 0.007). Sleeping time was associated with perceived stress score (*F* = 11.56; *p* < 0.001), anxiousness (*F* = 13.79; *p* < 0.001), and optimism (*F* = 4.92; *p* = 0.027). Daily sitting time was associated with all of the psychological well-being variables; perceived stress (*F* = 17.83; *p* < 0.001), quality of life (*F* = 30.49; *p* < 0.001), happiness (*F* = 30.57; *p* < 0.001), anxiousness (*F* = 7.92; *p* = 0.005) and optimism (*F* = 4.46; *p* = 0.035). Strong association was recorded between lockdown period and quality of life (*F* = 101.63; *p* < 0.001), followed by the association with happiness (*F* = 26.95; *p* < 0.001), optimism (*F* = 24.60; *p* < 0.001), and slight association with perceived stress score (*F* = 7.75; *p* = 0.005), and no association with anxiousness (Table 4).

## 4. Discussion

The most interesting findings of this study concerns sleeping habits of adolescents from Split, Croatia, during COVID-19 lockdown. The median sleep duration was up to one and a half hour longer during the lockdown period, resulting in 20% less students reporting to feel extremely tired and sleepy when waking up, as compared to pre-COVID-19 lockdown period. Actually, sleep duration during lockdown was similar to sleep duration on non-working days during pre-COVID period, possibly due to more relaxed daily schedule of online classes. Other studies also found increased sleeping hours during lockdown [6,32]. Nevertheless, a study performed in nursing students found that despite the longer time spent in bed, students’ sleep quality was lower during lockdown [33]. Similar findings regarding sleep quality were described in working adults and university students from Italy, who reported both going to bed and waking up later, but despite spending more time in bed they reported lower sleep quality [34]. We didn’t actually assess sleep quality in our study, but we could estimate it indirectly considering the reported feeling of students when waking up, if refreshed or sleepy and tired after a night of sleep. Thus, this indicator looked better during lockdown. It seems likely that during the pre-COVID lockdown period, students were trying to compensate lack of sleep accumulated during the working days (school days) in their free days. Schools in Croatia usually have an early start, at 8 o’clock in the morning. Consequently, during the pre-COVID-19 lockdown period the median sleep duration of 7 h during working days tended to increase to 9 h during non-working days. This reflected on a higher proportion of students who felt refreshed during weekends (55% vs. 8.5% on working days), but revealing that a staggering 91.5% of students usually felt tired or sleepy on school days pre-COVID-19. Taking into account the overall role and effect of adequate sleep duration and sleep quality on various health outcomes, it is crucial to emphasize the importance of sleep in all stages of life, and particularly so during adolescence. Studies have shown that getting enough sleep as well as good quality of sleep influences both physical and mental wellbeing, positively impacting risk of cardiovascular disease, diabetes, and obesity [35,36]. It has been estimated that among university students, prevalence of poor sleep quality is around 60% or even higher [37]. Sleepiness and irregular sleep schedules may also impact learning and memory abilities. Therefore, college students with early morning classes may not attain the last 1–2 REM (rapid eye movement) sleep periods, thus adversely affecting procedural memory [13]. Furthermore, another study showed that looking at more global functions, total sleep deprivation resulted in a decrease of cognitive tasks performance assessing inference, recognition of assumptions, and deduction [38]. Consistently, multiple studies have suggested that early school and college start times may contribute to inadequate sleep [39,40,41]. Furthermore, a study assessing the effect of just 30 min of daily class schedule delayed start (8:30 a.m. instead of 8:00 a.m.) resulted in a significant increase in sleep duration and decrease in day time sleepiness [42]. In light of these findings, our students would greatly benefit from postponing the beginning of teaching schedules, which could enhance their academic performance and reduce the risk of various adverse events later in life.

Besides differences in sleeping habits, this study also demonstrated some other changes in daily habits due to COVID-19 lockdown, including eating habits, screen time and psychological well-being. The impact of lockdown on food intake among our students was equivocal. Meal patterns remained stable, unlike in previous studies, which showed an increased number of main meals and snacking [5,43], explained by spending more time on cooking, trying new recipes and enjoying food without rushing during lockdown [5]. Even though we didn’t find differences in breakfast frequency, number of meals per day, snacking frequency or overall Mediterranean diet adherence, some differences have been recorded between the two study periods. For example, during COVID-19 lockdown period students had higher MD adherence to consumption of fruit, legumes, fish and sweets, while reported consumption of cereals, nuts, and dairy products was lower than in the pre-COVID lockdown period. A study performed among 10 to 19 years old adolescents also found an increase in fruit, vegetables and legumes consumption by 7.7%, 7.8% and 2.3%, respectively [19]. In addition, it was also reported that fast food intake was reduced [19]. Another study confirmed a drastic decrease of fast-food frequency consumption during COVID-19, with up to 82% adults reporting no fast food consumption [18]. There was also a significant increase in the percentage of participants who home cooked their main meal, but without differences in consumption of fruit and vegetables [18]. We described a positive change in dietary pattern, with 20–38% of students recording an increased intake of fruit and vegetables during lockdown, nevertheless similar percentage of students reported an increased intake of sweets and snacks. Additionally, weight change was frequently perceived during the lockdown, with 32% and 19% of students reporting weight loss and weight gain, respectively. Interestingly, students who reported major changes in their diet during lockdown, were mainly within the sub-group of international medical students. Although they followed the best overall dietary pattern before COVID-19 epidemic, they also reported the higher decrease in fruit and vegetables intake during lockdown (in 10% of students), while 34% increased their intake of sweets and snacks. These students were away from their families, while in the same time student restaurants were closed, and they had to take care of grocery shopping and cooking on their own. Our results highlight the need for intervention and provision of adequate assistance to these students, ideally by faculty management. This could include a number of support services that current and future international students could access, such as nutrition support and psychological counselling.

Obviously, the overall pattern of change as well as the changes within the specific student sub-groups is rather complex and warrants further investigations. Closer studies are needed to elucidate the reasons behind various observed changes, focusing on availability of foods, stress-related eating and other factors, affecting behavior such as media recommendations and science news. Following those information, particular attention has been paid to the intake of micronutrients, including vitamin C and D during COVID-19 pandemic [44,45], since increased intake of fruit and vegetables has been recommended by several authors, in order to support the immune system [46,47].

Healthy eating and food choices during adolescence are important for supporting growth and for preventing future health problems in adult life [48]. Young people following an unhealthy diet have higher odds of weight increase [49,50,51], with plethora of possible negative health outcomes. A study conducted among Turkish freshman university students aged between 18 and 22 years has shown that students with obesity had lower awareness of CVD risk factors than normal-weight ones, while subjects with a family member with CVD as well as non-smokers were more aware of the importance of dietary factors in CVD development [52]. Such insights should be taken into account during promotion of healthy lifestyle among young people.

Especially worrisome is our finding of low adherence to the MD among adolescents from the Mediterranean part of Croatia. This surely represents a lost opportunity, since Mediterranean diet has been repeatedly acknowledged to positively impact both physical health [53] and psychological well-being [54,55]. Our results are in agreement with those findings since we found a positive correlation between MD adherence score and the quality of life in our sample. Furthermore, MD adherence can be regarded as a proxy of a more healthy lifestyle since it was positively correlated with study time and optimism, and negatively with TV and mobile phone use and sleep duration during non-working days (oversleeping) in pre-lockdown study period. Additionally, students with higher MD adherence had lesser perception of hardship during lockdown, shorter mobile phone use (negative correlation for both associations), and greater happiness and quality of life during lockdown (positive correlation). Those results could be used to argue for placing MD as a focal point of lifestyle medicine application, since it is well known that “nutrition plays a key role in lifestyle habits and practices that affect virtually every chronic disease” [56]. In short, a new and rapidly developing field of lifestyle medicine, which is possibly heading towards becoming a new specialty, can be defined as the medical field that uses “lifestyle therapeutic approaches” to prevent, treat, or modify non-communicable chronic diseases [57].

Another important finding of this study refers to the time spent on computer/tablet, which increased in 70% of students and on average up to two hours during COVID-19 lockdown, mainly due to transition to online learning. Additionally, 63% of students reported an increase in the amount of time spent in mobile phones use, probably used also for school related activities. However, even if screen time is usually associated with sedentary behavior and snacking, we recorded less sitting time during COVID-19 lockdown compared to previous period. This may be due to the fact that students were not forced to sit for a fixed amount of time in classrooms, and screen time could have been used to encourage physical activity [58]. Indeed, various platforms such as online physical activity classes, exercise applications on mobile devices, or video games promote physical activity [59].

The results of studies conducted so far on the physical activity and COVID-19 lockdown relationship are contradictory, showing either an increase of sedentariness [6,51,60,61,62], or a decrease [63]. Our results indicated that 41% of students surveyed in 2020 were active a few times a week, while additional 24% were active every day during lockdown. Those are satisfactory percentages, confirming a good amount of physical activity carried out both before and during the epidemic. Exercise supports not only physical health, but also mental well-being [64]. Another study suggested that regular physical activity may help children and adolescents to recover from stress and anxiety they experience during lockdown [65]. While scientists struggle to find an optimized solution to control SARS-CoV-2 virus spread, promoting exercise can be a good strategy not only for weight control [8], but it can also be implemented for improving mental health and functioning of the immune system [66].

Another important issue that needs to be addressed is COVID-19 lockdown impact on mental health and well-being. Our subjects reported slightly lower average quality of life during COVID-19 lockdown compared to pre-COVID-19 period, and a slight increase in stress perception. A decline in subjective well-being during COVID-19 pandemic, and stress and anxiety increase in adolescents have also been reported in previous studies [10,21]. This is quite understandable, considering the magnitude of changes in daily life due to lockdown experience. These changes include reorganization of family life, increased stress, fear of death of elderly relatives, limited access to health services, and lack of social stabilization from peers, teachers and sport activities, which may lead to dangerous accumulation of risk factors for mental health problems in children and adolescents [67]. In a rapid review of the evidence recently published on quarantine psychological impact, it was revealed that longer quarantine duration, infection fears, frustration, boredom, inadequate supplies, inadequate information, financial loss, and stigma were the most prominent stressors [9]. Despite all those challenges and possible threats to mental health, our subjects reported a medium level of hardship experience due to lockdown and isolation, and overall high self-rated health perception in all study groups. This may be due to appropriate coping strategies students have employed, such as longer sleep duration during lockdown, regular physical activity, and increased consumption of fruit and vegetables. Additionally, great efforts were made to maintain learning continuity during this period to face COVID-19 lockdown in Croatia. Ministry of Science and Education has introduced the online school system rapidly at the exact moment when lockdown started, via national TV broadcasting and online learning platforms. Nevertheless, lockdown affected quality of life, perceived stress and happiness, and optimism about future in secondary school students and medical students from Split, Croatia. This underlines the need for further surveillance of psychological well-being in these people and screening for disturbances in mental health during times of great disruption and major change.

Several limitations of this study should be considered. They include a cross-sectional design and different subjects enrolled in each school and university year (we did not use the follow-up approach in the same students). Additionally, the students were asked to recall their behaviors, possibly introducing a recall bias. Still, we assume that this type of bias had a small impact on our results due to the young age of our subjects and the questions being generally focused on the period of one month prior to data collection. Furthermore, we used a different data collection approach during two study periods. Namely, we used a paper-based approach in pre-COVID-19 period and a web-based approach during COVID-19 lockdown. This might have led to uncertain amount of bias. However, one previous study performed in a similar population of secondary school students, found that a questionnaire focused on general health, well-being and behavior health indicators yielded almost the same results when administered either by paper-and-pencil or by web [68]. Moreover, even though authors who developed the PSS-10 questionnaire did not propose dichotomization of the score [30], we have applied one of the previously proposed cut-off values to categorize students into groups of low, moderate or high level of perceived stress. This is an intuitive approach in defining stress perception and it could enable comparison of our result with other similar studies. However, we did not use categorical variable in regression analysis, but instead we used numerical PSS-10 score.

Finally, height and weight were self-reported, which might not be very precise or accurate approach in assessing subjects’ nutritional status. It was previously recognized that self-reported weight data underestimate overweight prevalence and that bias by sex and weight status exists, but they do provide valuable information if they are the only source of data [69]. However, we did ask students a control question about how many days ago students had weighted themselves, and only 33% of students responded that they weighted themselves more than 30 days ago.

The strengths of this study include relatively large sample size, high response rates which increase representativeness of the results, and inclusion of three similar, yet different subgroups of students. There were some variations in response rates among medical students, where international students had the lowest response rate in pre-COVID-19 period (58.1% in 2019 and 62.8% in 2018), and domestic students had the lowest response rates in lockdown period (58.4%). This could be a random variation, since we have had the same approach to data collection in all study groups.

Furthermore, we provided the elaborate description of adolescents’ lifestyle, with many important characteristics and behaviors in the domain of nutrition, physical activity, and sleep. Finally, we investigated the association of these habits with the psychological well-being in adolescents, an important and vulnerable sub-group of population going through a turbulent period of life as it is, hurled into the hectic and enormously challenging period of COVID-19 pandemic.

## 5. Conclusions

In conclusion, this study provides an overview of the changes in dietary habits, sleep, physical activity, and psychological well-being due to COVID-19 lockdown in youth, including adolescents and medical students. The epidemic brought various changes in everyday life and habits, some beneficial and some detrimental. Still, there is much to be investigated considering that we have estimated only short-term consequences of lockdown on adolescents’ and young adults’ lifestyle. Given the uncertain course of COVID-19 pandemic in months and years to come, these insights provide valuable information for tailored interventions aimed at maintaining healthy lifestyle in young population and hence preserving their health and managing stress. Such information is important for public health authorities, but also for educational institutions, which should consider responding to the physical and mental health needs of students during the COVID-19 pandemic and in relation to future uncertainty [70]. The main focus should be on maintaining the well-being of students for better productivity and lower burden of negative consequences, and providing young people with more opportunities to lead fulfilling lives as adults.

## Figures and Tables

**Figure 1 nutrients-13-00097-f001:**
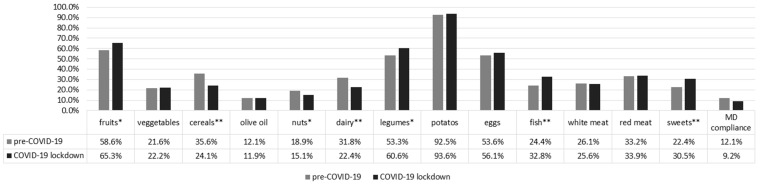
Adherence to the Mediterranean diet pyramid food groups according to the Mediterranean diet score, excluding wine due to subjects’ age [24] and the overall Mediterranean diet (MD) adherence (** *p* < 0.001, * *p* < 0.05).

**Table 1 nutrients-13-00097-t001:** General characteristics of the sample before and during COVID-19 lockdown.

	Pre-COVID-19 (2018 and 2019)				COVID-19 Lockdown			
	Secondary School Students *n* = 769	Domestic Medical Students *n* = 413	International Medical Students *n* = 144	*p*	Secondary School Students *n* = 324	Domestic Medical Students *n* = 148	International Medical Students *n* = 59	*p*
Age; median (IQR)	17.0 (1.0)	23.0 (5.0)	24.0 (4.0)	<0.001 ^#^	17.5 (1.0)	23.0 (2.0)	22.0 (6.0)	<0.001 ^#^
Gender; *N* (%)				0.046 *				0.084 *
Females	470 (61.5)	283 (68.7)	89 (62.2)		232 (71.6)	115 (77.7)	37 (62.7)	
Males	294 (38.5)	129 (31.3)	54 (37.8)		92 (28.4)	33 (22.3)	22 (37.3)	
Smoking; *N* (%)				0.564 *				0.212 *
yes	113 (14.9)	72 (17.5)	25 (17.4)		46 (14.2)	24 (16.2)	15 (25.4)	
ex-smokers	78 (10.3)	43 (10.5)	19 (13.2)		32 (9.9)	19 (12.8)	6 (10.2)	
never smoked	568 (74.8)	296 (72.0)	100 (69.4)		246 (75.9)	105 (70.9)	38 (64.4)	
BMI; median (IQR)	21.0 (2.9)	21.8 (3.5)	22.7 (3.5)	<0.001 ^#^	21.3 (2.9)	22.0 (3.7)	22.0 (3.4)	0.015 ^#^
Weighing (days ago); *N* (%)				0.006 *				0.029 *
within 7 days	258 (37.7)	78 (38.6)	41 (31.3)		177 (54.6)	78 (52.7)	32 (54.2)	
8–30 days ago	204 (29.8)	81 (40.1)	42 (32.1)		84 (25.9)	54 (36.5)	13 (22.0)	
≥31 days ago	222 (32.5)	43 (21.3)	48 (36.6)		63 (19.4)	16 (10.8)	14 (23.7)	
Self-rated health perception; median (IQR)	9.0 (1.0)	9.0 (2.0)	9.0 (2.0)	0.172 ^#^	9.0 (2.0)	9.0 (2.0)	8.0 (2.0)	0.001 ^#^

IQR—interquartile range; * chi-square test, ^#^ Kruskal-Wallis test.

**Table 2 nutrients-13-00097-t002:** Lifestyle habits before and during COVID-19 lockdown in the overall sample.

	Pre-COVID-19 (2018 and 2019) *n* = 1326	COVID-19 Lockdown *n* = 531	*p*
Age; median (IQR)	18.0 (6.0)	18.0 (6.0)	0.100 ^§^
Gender; *N* (%)			<0.001 *
Females	842 (63.8)	384 (72.3)	
Males	477 (36.2)	147 (27.7)	
BMI; median (IQR)	21.4 (3.3)	21.5 (3.2)	0.563 ^§^
Self-rated health perception; median (IQR)	9.0 (1.0)	9.0 (2.0)	0.001 ^§^
**Dietary habits**			
Breakfast frequency; *N* (%)			0.350 *
every day	757 (57.4)	324 (61.0)	
4–6 days/week	274 (20.8)	99 (18.6)	
0–3 days/week	288 (21.8)	108 (20.3)	
Snacking while watching TV or studying ^†^; *N* (%)			0.341 *
yes, frequently	115 (15.7)	79 (14.9)	
yes, sometimes	418 (57.0)	287 (54.0)	
no	200 (27.3)	165 (31.1)	
MDSS score; median (IQR)	7.0 (6.0)	7.0 (5.0)	0.927 ^§^
**Sleeping habits**			
Sleep duration on working days; median (IQR)	7.0 (1.5)	8.5 (1.5)	<0.001 ^§^
Feeling after waking up on working days; N (%)			<0.001 *
refreshed	112 (8.5)	167 (31.5)	
somewhat tired and sleepy	804 (60.8)	310 (58.4)	
extremely tired and sleepy	407 (30.8)	54 (10.2)	
Sleep duration on non-working days; median (IQR)	9.0 (1.5)	8.5 (1.5)	<0.001 ^§^
Feeling after waking up on non-working days; *N* (%)			<0.001 *
refreshed	722 (54.7)	167 (31.5)	
somewhat tired and sleepy	531 (40.3)	310 (58.4)	
extremely tired and sleepy	66 (5.0)	54 (10.2)	
**Sedentary activity**			
Sitting time (h/day); median (IQR)	7.0 (4.0)	6.0 (5.0)	<0.001 ^§^
TV watching time (h/day); median (IQR)	1.0 (1.5)	1.0 (1.8)	<0.001 ^§^
Computer/tablet use time (h/day); median (IQR)	1.0 (2.0)	3.0 (3.5)	<0.001 ^§^
Mobile use time (h/day); median (IQR)	3.0 (3.0)	3.0 (3.0)	0.247 ^§^
Studying time (h/day); median (IQR)	3.0 (2.5)	3.0 (3.0)	<0.001 ^§^
**Psychological well-being**			
Happiness; median (IQR)	7.0 (2.0)	7.0 (3.0)	<0.001 ^§^
Optimistic about future; median (IQR)	7.0 (4.0)	7.0 (3.0)	<0.001 ^§^
Anxiousness; median (IQR)	4.0 (5.0)	4.0 (5.0)	0.010 ^§^
Perceived stress score (PSS-10); median (IQR)	18.0 (11.0)	19.0 (10.0)	0.037 ^§^
Perceived stress category; *N* (%)			0.101 *
low	340 (25.7)	112 (21.1)	
moderate	778 (58.9)	336 (63.3)	
high	203 (15.4)	83 (15.6)	
Quality of life; median (IQR)	8.0 (2.0)	7.0 (2.0)	<0.001 ^§^

IQR—interquartile range, MDSS—Mediterranean Diet Serving Score; * chi-square test, ^§^ Mann-Whitney U test, ^†^ data available only for 2019 for pre-COVID study period (sample size is 733), during COVID-19 study period only one set of sleep data was collected, and we compared it with both sets of sleep data from pre-COVID-19 study period (for working days and non-working days).

**Table 3 nutrients-13-00097-t003:** Self-perceived habits change in students during COVID-19 lockdown.

	Total *n* = 531	Secondary School Students *n* = 324	Domestic Medical Students *n* = 148	International Medical Students *n* = 59	*p*
Fruits; *N* (%)					0.507 *
reduced	29 (5.5)	17 (5.2)	6 (4.1)	6 (10.2)	
remained the same	319 (60.1)	195 (60.2)	89 (60.1)	35 (59.3)	
increased	183 (34.5)	112 (34.6)	53 (35.8)	18 (30.5)	
Vegetables; *N* (%)					<0.001 *
reduced	17 (3.2)	9 (2.8)	2 (1.4)	6 (10.2)	
remained the same	383 (72.1)	255 (78.7)	89 (60.1)	39 (66.1)	
increased	131 (24.7)	60 (18.5)	57 (38.5)	14 (23.7)	
Meat and processed meat; *N* (%)					0.316 *
reduced	45 (8.5)	25 (7.7)	11 (7.4)	9 (15.3)	
remained the same	422 (79.5)	262 (80.9)	116 (78.4)	44 (74.6)	
increased	64 (12.1)	37 (11.4)	21 (14.2)	6 (10.2)	
Sweets and snacks; *N* (%)					0.170 *
reduced	143 (26.9)	97 (29.9)	31 (20.9)	15 (25.4)	
remained the same	243 (45.8)	141 (43.5)	78 (52.7)	24 (40.7)	
increased	145 (27.3)	86 (26.5)	39 (26.4)	20 (33.9)	
Weight change; *N* (%)					0.002 *
reduced	167 (31.5)	96 (29.6)	46 (31.1)	25 (42.4)	
remained the same	198 (37.3)	118 (36.4)	58 (39.2)	22 (37.3)	
increased	102 (19.2)	56 (17.3)	35 (23.6)	11 (18.6)	
don’t know	64 (12.1)	54 (16.7)	9 (6.1)	1 (1.7)	
TV watching time; *N* (%)					<0.001 *
reduced	73 (13.7)	45 (13.9)	14 (9.5)	14 (23.7)	
remained the same	339 (63.8)	226 (69.8)	85 (57.4)	28 (47.5)	
increased	119 (22.4)	53 (16.4)	49 (33.1)	17 (28.8)	
Computer/tablet use time; *N* (%)					<0.001 *
reduced	21 (4.0)	12 (3.7)	8 (5.4)	1 (1.7)	
remained the same	137 (25.8)	58 (17.9)	57 (38.5)	22 (37.3)	
increased	373 (70.2)	254 (78.4)	83 (56.1)	36 (61.0)	
Mobile use time; *N* (%)					0.001 *
reduced	28 (5.3)	13 (4.0)	8 (5.4)	7 (11.9)	
remained the same	171 (32.2)	90 (27.8)	54 (36.5)	27 (45.8)	
increased	332 (62.5)	221 (68.2)	86 (58.1)	25 (42.4)	
Studying time; *N* (%)					0.020 *
reduced	172 (32.4)	102 (31.5)	57 (38.5)	13 (22.0)	
remained the same	200 (37.7)	119 (36.7)	60 (40.5)	21 (35.6)	
increased	159 (29.9)	103 (31.8)	31 (20.9)	25 (42.4)	
Physical activity; *N* (%)					0.173 *
active every day of the week	125 (23.5)	87 (26.9)	30 (20.3)	8 (13.6)	
active a few days a week	217 (40.9)	129 (39.8)	59 (39.9)	29 (49.2)	
sometimes moderately active	82 (15.4)	50 (15.4)	21 (14.2)	11 (18.6)	
very low level of activity and rarely active	77 (14.5)	45 (13.9)	25 (16.9)	7 (11.9)	
not physically active at all	30 (5.6)	13 (4.0)	13 (8.8)	4 (6.8)	
How hard has it been to experience lockdown and to be isolated from others?; median (IQR)	6.0 (5.0)	6.0 (5.0)	6.0 (5.0)	5.0 (6.0)	0.623 ^#^

IQR—interquartile range; * chi-square test, ^#^ Kruskal-Wallis test.

**Table 4 nutrients-13-00097-t004:** Association of the Mediterranean diet, sleep duration, sedentary activity, and COVID-19 lockdown with psychological well-being in students, using multivariate general linear model (all psychological well-being variables were simultaneously entered as dependent variables).

Predictors	Outcomes (Dependent Variables)	*F*	*p*	Partial η^2^
Gender (males are referent group)	Perceived stress score (PSS-10)	73.62	<0.001	0.070
	Quality of life	5.51	0.019	
	Happiness	6.89	0.009	
	Anxiousness	39.30	<0.001	
	Optimistic about future	23.22	<0.001	
Age (years)	Perceived stress score (PSS-10)	6.47	0.011	0.007
	Quality of life	3.00	0.084	
	Happiness	4.84	0.028	
	Anxiousness	1.99	0.159	
	Optimistic about future	1.08	0.299	
Study group (secondary school students are referent group)	Perceived stress score (PSS-10)	7.76	<0.001	0.008
	Quality of life	3.90	0.020	
	Happiness	4.56	0.011	
	Anxiousness	2.55	0.078	
	Optimistic about future	1.28	0.278	
MDSS score	Perceived stress score (PSS-10)	1.63	0.202	0.017
	Quality of life	14.95	<0.001	
	Happiness	1.70	0.193	
	Anxiousness	0.52	0.472	
	Optimistic about future	7.23	0.007	
Working days sleeping time (h)	Perceived stress score (PSS-10)	11.56	0.001	0.015
	Quality of life	0.44	0.506	
	Happiness	3.35	0.068	
	Anxiousness	13.79	<0.001	
	Optimistic about future	4.92	0.027	
Sitting time daily (h)	Perceived stress score (PSS-10)	17.83	<0.001	0.034
	Quality of life	30.49	<0.001	
	Happiness	30.57	<0.001	
	Anxiousness	7.92	0.005	
	Optimistic about future	4.46	0.035	
COVID-19 lockdown (pre-COVID-19 is referent group)	Perceived stress score (PSS-10)	7.75	0.005	0.089
	Quality of life	101.63	<0.001	
	Happiness	26.95	<0.001	
	Anxiousness	1.09	0.297	
	Optimistic about future	24.60	<0.001	

## Data Availability

The data presented in this study are available on request from the corresponding author.

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
