# Peer review of "Have Lifestyle Habits and Psychological Well-Being Changed among Adolescents and Medical Students Due to COVID-19 Lockdown in Croatia?"

_nutrients, 2020, doi:10.3390/nu13010097_

Round 1

Reviewer 1 Report

Thank you for the opportunity to review the manuscript “Have lifestyle habits and psychological well-being 2 changed among adolescents and medical students 3 due to COVID-19 lockdown in Croatia?”. Situations such as a pandemic and the related actions of public health authorities to limit its spread may and often have a significant impact on lifestyle habits and psychological well-being. Therefore, the task undertaken by the authors of the manuscript is very important for the assessment of changes in these behaviors during the pandemic that affected all of Europe, including Croatia.

Matherials and tethods

What was the reason for using a Likert scale of 1-10? Usually it is used as 1-5 or 1-7.

It was a good idea to make a model illustrating the effect of a Mediterranean Diet, sleep duration, sedentary activity, and COVID-19 lockdown on "overall psychological well-being".

However, the multivariate general linear model used was not vaguely described. What was the selection of variables for the model? What level of statistical significance was adopted (I understand 0.05)? Please also describe the influence of the obtained effects of the model on the independent variable.

Author Response

  1. Thank you for the opportunity to review the manuscript “Have lifestyle habits and psychological well-being 2 changed among adolescents and medical students 3 due to COVID-19 lockdown in Croatia?”. Situations such as a pandemic and the related actions of public health authorities to limit its spread may and often have a significant impact on lifestyle habits and psychological well-being. Therefore, the task undertaken by the authors of the manuscript is very important for the assessment of changes in these behaviors during the pandemic that affected all of Europe, including Croatia.

Answer: Thank you for your response and valuable comments.

  1. Matherials and methods

 What was the reason for using a Likert scale of 1-10? Usually it is used as 1-5 or 1-7.

Answer:We acknowledge that the most commonly used Likert scale is the one with 5 or 7 points. However, we have decided to apply the 10-point scale, in order to provide a greater range of possible answers to our subjects, especially since we used this type of question for sensitive, psychological characteristics, such as anxiety, happiness, optimism, quality of life and self-rated health perception. Indeed, it was shown that: „A 5- or 7-point scale is likely to produce slightly higher mean scores relative to the highest possible attainable score, compared to that produced from a 10-point scale.“ (Dawes J. Do data characteristics change according to the number of scale points used? An experiment using 5-point, 7-point and 10-point scales. Int. J. Mark. Res.2008;50:61-77.) Therefore, the main reason we used a 10-point scale was to offer a broader possibility of the answers to students regarding their feelings. This was already found previously: “On the other hand, respondents felt that the scales with more options (nine-, 10-, 11-, and 101-point scales) allowed them greater expression of feelings. The researchers concluded that overall, the 10-point scale scored best, followed by the seven-point and nine-point scales.“ (https://core.ac.uk/download/pdf/147014152.pdf)

Furthermore, a common grading system in schools and universities in Croatia is the use of grades from 1 to five, where 1 represents failure, and 5 corresponds to excellent. We wanted to step away from this scale, in order to detach self-rating of students’ psychological well-being from grades system. Finally, we have used a 10-point scale in our previous studies and in our subsequent data collection, which will make the results comparable in a straightforward way between this and our next publications.

  1. It was a good idea to make a model illustrating the effect of a Mediterranean Diet, sleep duration, sedentary activity, and COVID-19 lockdown on "overall psychological well-being".

However, the multivariate general linear model used was not vaguely described. What was the selection of variables for the model? What level of statistical significance was adopted (I understand 0.05)? Please also describe the influence of the obtained effects of the model on the independent variable.

Answer:Thank you for this comment and questions.As a dependent variable, we decided to use all of the variables related to psychological wellbeing: perceived stress, quality of life, happiness, anxiousness, and optimism about future. In order to do that, we chose this particular approach of multivariate analysis - general linear model, where we could enter all of these variables at once. As predictor variables we used the most important lifestyle factors we have included in our study (Mediterranean diet, sleeping time, and sitting time as a measure of sedentary activity), alongside with study period (pre-COVID-19 was referent group). Age, gender, and study group (type of students) were regarded as confounding factors. We have added this description in the statistical section (page 5, lines 229-238).

Yes, we have uniformly applied P<0.05 as a significance cut-off point.

We have also described the influence of the obtained effects of the model, by adding the partial eta squared values to Table 4, in order to report the estimates of effect sizes (page 14). Those values indicate the percentage of variance in each of the effects and its associated error that is accounted for by that effect. Additionally, we reported adjusted R2 values (page 5, lines 239-243).

Reviewer 2 Report

General comments:

The work reported in this paper addresses an important issue of lifestyle changes and its impact during the COVID pandemic among a crucial and vulnerable group. This work is timely, to explore and understand the behaviours patterns to support the young people during such pandemics.  However, I have few minor comments

Introduction and background:  

  • Introduction is quite long. It could be made a bit more succinct.

Methods:

  • They have included 3 schools out of 6, how were the 3 school’s selected? (pg 3; 110)
  • Authors say two generations of secondary school students were involved. Not clear what the term ‘generation’ meant. Do they mean two consecutive years?? This needs to be made clear. (pg 3; 110)
  • How was the sample size calculated and how were the young people recruited?
  • For the second study period (post COVID) the total sample is 534 (326+149+59) (Pg 3; lines 124-125)….however in the abstract and results (pg5; line230) and Tables, it says 531. Please check
  • Page 4, line 155, please expand MDSS
  • Pg 5, line 198, authors said that the PSS scores were dichotomised, and it is not usually approach, what are the implications of this approach?

Results:

  • Results are presented clearly….however worth mentioning that the self-rated health perception is lower in the international medical students post covid, which seems to be statistically significant (p 0.001) (Table 1)
  • What was the reference group for age? (Table 4)

Discussion:

  • In the pre- covid survey, the international medical student’s response rate was quite low compared to domestic students (pg 3; 191-121). In the post-covid survey the domestic students were the lowest (pg 3; 124-25). It is a very interesting observation. Any explanations for that??
  • Authors say that “In order to increase the response rate in all instances of data collection, we asked professors for their help in the questionnaire distribution during their regular teaching hours” (pg3; 141-153). While I see conducting the survey during the class will increase response rate, there is a potential that this could have introduced selection bias or performance bias. Some discussion on this would be good.
  • Height and Weight are self-reported, some discussion on the validity of self -reporting would add strength to the paper.
  • Major changes in diet (unhealthy) was among international medical students. Authors warrant further investigation, however, some discussion on this will be good as to how they can be helped.

Page 15, line 422, tiny typo: ‘your’ should be changed to ‘young’

Author Response

The work reported in this paper addresses an important issue of lifestyle changes and its impact during the COVID pandemic among a crucial and vulnerable group. This work is timely, to explore and understand the behaviours patterns to support the young people during such pandemics.  However, I have few minor comments

 Introduction and background:  

  • Introduction is quite long. It could be made a bit more succinct.
  • Answer: As suggested, we have shorten our Introduction and now it has 759 words in total (page 2). However, we kept a short description of the epidemiologic measures and lockdown situation in Croatia; because we think that it is relevant for readers to get a glimpse of the experience of the lockdown, since we investigate its impact on lifestyle habits and psychological well-being on young people in our paper.

 Methods:

  • They have included 3 schools out of 6, how were the 3 school’s selected? (pg 3; 110)
  • Answer: All of the schools were invited and five of them participated in the pre-COVID-19 period in data collection. However, during lockdown period only three schools have been able to provide online response by students for their various reasons. This is the reason why we included only those three schools in our analysis, since we had both data before and during lockdown period for those schools, enabling direct comparisons. We have added more information about this in Materials and Methods (page 3, lines 114-118).
  • Authors say two generations of secondary school students were involved. Not clear what the term ‘generation’ meant. Do they mean two consecutive years?? This needs to be made clear. (pg 3; 110)
  • Answer: Yes, we meant to say students enrolled in two consecutive academic years. We have made this clearer (page 3, lines 110-111).
  • How was the sample size calculated and how were the young people recruited?
  • Answer: We have invited the whole generations to participate, aiming to includeall of the students within generations of students in 2018, 2019 and 2020, for all three groups of students.We have added this info on page 3, lines 109-110, and line 125. The reason we did not perform sample size calculation is that, technically, we were not performing sampling.Of course, we aimed to include all students enrolled (the whole target population), but we did not succeed entirely. However, we believe that response rates of 89.5% (in 2018), 93.6% (in 2019) and 77.1% (in 2020) among secondary school studentsare large enough to represent the whole target population, while the response rates among medical students were somewhat lower, but never lower than 58%.
  • Recruitment of students was done with the help of secondary school professors (their names are included in the Acknowledgments; they also helped with obtaining of signed informed consent by parents for students who were younger than 18 yrs). Among medical students, we asked their students’ representatives and their professors for their help in distributing paper-based questionnaire and the link for the online survey (added on page 4, line 148).
  • For the second study period (post COVID) the total sample is 534 (326+149+59) (Pg 3; lines 124-125)….however in the abstract and results (pg5; line230) and Tables, it says 531. Please check
  • Answer: Thank you for this observation. We had to exclude one domestic medical student and two secondary school students from the analysis due to incomplete questionnaire response. Somehow, we missed to point this out in our submitted version of the manuscript. We have added this explanationnow in the beginning of the Results section (on page 5-6, lines246-248).
  • Page 4, line 155, please expand MDSS
  • Answer: Thank you, we corrected this (now page 4, line 163)
  • Pg 5, line 198, authors said that the PSS scores were dichotomised, and it is not usually approach, what are the implications of this approach?

Answer: This was added to the limitations section: “Moreover, even though authors who developed the PSS-10 questionnaire did not propose dichotomization of the score [30], we have applied one of the previously proposed cut-off values to categorize students into groups of low, moderate or high level of perceived stress. This is an intuitive approach in defining stress perception and it could enable comparison of our result with other similar studies. However, we did not use categorical variable in regression analysis, but instead we used numerical PSS-10 score.” (page , lines 515-521).

 Results:

  • Results are presented clearly….however worth mentioning that the self-rated health perception is lower in the international medical students post covid, which seems to be statistically significant (p 0.001) (Table 1)

Answer: Thank you for pointing this out, we have added the following text to the previous sentence: “Self-rated health perception was high in all study groups, while statistically significantly lower in international medical students during COVID-19 lockdown (median 8.0, IQR 2.0), compared to domestic medical students (median 9.0, IQR 2.0) and secondary school students (median 9.0, IQR 2.0; P<0.001) (Table 1).” (page 6, lines 256-259).

  • What was the reference group for age? (Table 4)
  • Answer: We have added more information and description of the regression model, stating which variables were entered as numerical variables, such as age, Mediterranean diet, sleep duration, and sitting time, and which variables were categorical:

“We included the most important lifestyle habits as predictor variables in the model (Mediterranean diet, sleep duration, and sitting time as a measure of sedentary type of physical activity), alongside with time period (pre-COVID-19 was the referent point). All of these variables, except study period, were entered into the model as numerical variables. The model was also adjusted for the effect of confounding variables; age (numerical variable), gender (males were the referent group), and study group (secondary school students were the referent group).”(page 5, lines 233-239).

Discussion:

  • In the pre- covid survey, the international medical student’s response rate was quite low compared to domestic students (pg 3; 191-121). In the post-covid survey the domestic students were the lowest (pg 3; 124-25). It is a very interesting observation. Any explanations for that??
  • Answer: This is indeed interesting. This could be a random variation, since we have had the same approach to all study groups. We have added this sentence to limitations: “There were some variations in response rates among medical students, where international students had the lowest response rate in pre-COVID-19 period (58.1% in 2019 and 62.8% in 2018), and domestic students had the lowest response rates in lockdown period (58.4%). This could be a random variation, since we have had the same approach to data collection in all study groups.” (page 18, lines 533-536).
  • Authors say that “In order to increase the response rate in all instances of data collection, we asked professors for their help in the questionnaire distribution during their regular teaching hours” (pg3; 141-153). While I see conducting the survey during the class will increase response rate, there is a potential that this could have introduced selection bias or performance bias. Some discussion on this would be good.
  • Answer: Thank you very much for pointing this out. We have not stated it clearly enough, and we see that our sentence structure can have a misleading effect. We did not ask professors to ask students to respond to the survey during the class, but instead professors have informed students about this survey and asked them to fill the questionnaire at their earliest convenience and to return the paper-based questionnaires to students who were at the time helping with data collection (first five authors of this study). During online teaching in lockdown study period, professors shared the link for online survey among students attending their obligatory classes (the whole generation was attending as a default). We added this explanation on page 4, lines 148-151.
  • Height and Weight are self-reported, some discussion on the validity of self -reporting would add strength to the paper.
  • Answer: We added this into limitations section, along with one new reference (please see page 17, lines 525-530):

“Finally, height and weight were self-reported, which might not be very precise or accurate approach in assessing subjects’ nutritional status. It was previously recognized that self-reported weight data underestimate overweight prevalence and that bias by sex and weight status exists, but they do provide valuable information if they are the only source of data [69]. However, we did ask students a control question about how many days ago students had weighted themselves, and only 33% of students responded that they weighted themselves more than 30 days ago.”

  • Major changes in diet (unhealthy) was among international medical students. Authors warrant further investigation, however, some discussion on this will be good as to how they can be helped.
  • Answer: Thank you very much for this observation.We have added a few thought regarding this comment (please see page 16, lines 429-433):

“These students were away from their families, while in the same time student restaurants were closed, and they had to take care of grocery shopping and cooking on their own. Our results highlight the need for intervention and provision of adequate assistance to these students, ideally by faculty management. This could include a number of support services that current and future international students could access, such as nutrition support and psychological counselling.”

  • Page 15, line 422, tiny typo: ‘your’ should be changed to ‘young’
  • Answer: We have amended this, thank you for observing this mistake.

Reviewer 3 Report

This paper studied dietary habits and other related lifestyle behaviours changes in students resulting to COVID-19 lockdown in Croatia. In particular, authors showed results about three groups (secondary school students, Croatian medical students and international medical students) of students, during two study periods (April-May 2018-19 with a  paper-and-pencil form and May 2020 with an online data collection approach). A total of 1326 students sampled during pre-COVID-19 study period and 531 students during COVID-19 lockdown period were included in the analysis. Authors provided an anonymous self-administered questionnaire, concerning age, gender, school/study details, smoking habits, student’s weight,  height and health perception, dietary habits (Mediterranean diet adherence), sleeping habits, sedentary types of activity, psychological wellbeing (perceived stress, quality of life, happiness, anxiety, and optimism about future).

I’m totally agree with the authors about the importance to “provide valuable information for tailored interventions aimed at maintaining healthy lifestyle in young population and hence preserving their health and managing stress”. Honestly, the reference to "lifestyle therapeutic approaches” seems to be of considerable clinical and public interest; the authors provide ample contribution through the results presented in this article.

The main, interesting, research question seems to be original, also considering that the study showed results relative to the Covid-19 emergence period and the previous. Moreover, the authors give an advance in current knowledge about research topic (“…only few studies have been published so far about COVID-19 lockdown influence on lifestyle change in adolescents  and in medical students”).

In my honest opinion, ethics aspect, as well as methodological and statistical aspects, are well explained.

Results are detailed, enriched by many tables and one very clear figure.

I fully agree with what the authors reported in discussion section about work limitations. 

The paper conclusions are shareable and in line with the results.

Author Response

This paper studied dietary habits and other related lifestyle behaviours changes in students resulting to COVID-19 lockdown in Croatia. In particular, authors showed results about three groups (secondary school students, Croatian medical students and international medical students) of students, during two study periods (April-May 2018-19 with a  paper-and-pencil form and May 2020 with an online data collection approach). A total of 1326 students sampled during pre-COVID-19 study period and 531 students during COVID-19 lockdown period were included in the analysis. Authors provided an anonymous self-administered questionnaire, concerning age, gender, school/study details, smoking habits, student’s weight,  height and health perception, dietary habits (Mediterranean diet adherence), sleeping habits, sedentary types of activity, psychological wellbeing (perceived stress, quality of life, happiness, anxiety, and optimism about future).

I’m totally agree with the authors about the importance to “provide valuable information for tailored interventions aimed at maintaining healthy lifestyle in young population and hence preserving their health and managing stress”. Honestly, the reference to "lifestyle therapeutic approaches” seems to be of considerable clinical and public interest; the authors provide ample contribution through the results presented in this article.

The main, interesting, research question seems to be original, also considering that the study showed results relative to the Covid-19 emergence period and the previous. Moreover, the authors give an advance in current knowledge about research topic (“…only few studies have been published so far about COVID-19 lockdown influence on lifestyle change in adolescents  and in medical students”).

In my honest opinion, ethics aspect, as well as methodological and statistical aspects, are well explained.

Results are detailed, enriched by many tables and one very clear figure.

I fully agree with what the authors reported in discussion section about work limitations. 

The paper conclusions are shareable and in line with the results.

  • Answer:

Thank you very much for your valuable input about our paper, we strongly believe that such studies are indeed needed in the general population, since we are facing unprecedented threat to human health and well-being, with very uncertain future regarding the resolution of this pandemic. We are happy to contribute to the baseline global knowledge with our results, about the influence of lockdown, which is the strictest approach among epidemiologic measures employed so far. We think that adolescents and young adults might be a sub-group of the population with the most detrimental long-term psychological effects of the COVID-19 pandemic.